# Concurrent Validity of Four Activity Monitors in Older Adults

**DOI:** 10.3390/s24030895

**Published:** 2024-01-30

**Authors:** Jorgen A. Wullems, Sabine M. P. Verschueren, Hans Degens, Christopher I. Morse, Gladys L. Onambélé-Pearson

**Affiliations:** 1Department of Sport and Exercise Sciences, Institute of Sport, Faculty of Science and Engineering, Manchester Metropolitan University, Manchester M1 7EL, UK; jorgen.wullems@rabobank.nl (J.A.W.); c.morse@mmu.ac.uk (C.I.M.); 2Musculoskeletal Rehabilitation Research Group, Department of Rehabilitation Sciences, KU Leuven, 3001 Leuven, Belgium; sabine.verschueren@kuleuven.be; 3Department of Life Sciences, Institute of Sport, Faculty of Science and Engineering, Manchester Metropolitan University, Manchester M1 5GD, UK; h.degens@mmu.ac.uk; 4Institute of Sport Science and Innovations, Lithuanian Sports University, 44221 Kaunas, Lithuania

**Keywords:** ageing, accelerometry, physical behaviour

## Abstract

Sedentary behaviour (SB) and physical activity (PA) have been shown to be independent modulators of healthy ageing. We thus investigated the impact of activity monitor placement on the accuracy of detecting SB and PA in older adults, as well as a novel random forest algorithm trained on data from older persons. Four monitor types (ActiGraph wGT3X-BT, ActivPAL3c VT, GENEActiv Original, and DynaPort MM+) were simultaneously worn on five anatomical sites during ten different activities by a sample of twenty older adults (70.0 (12.0) years; 10 women). The results indicated that collecting metabolic equivalent (MET) data for 60 s provided the most representative results, minimising variability. In addition, thigh-worn monitors, including ActivPAL, Random Forest, and Sedentary Sphere—Thigh, exhibited superior performance in classifying SB, with balanced accuracies ≥ 94.2%. Other monitors, such as ActiGraph, DynaPort MM+, and GENEActiv Sedentary Sphere—Wrist, demonstrated lower performance. ActivPAL and GENEActiv Random Forest outperformed other monitors in participant-specific balanced accuracies for SB classification. Only thigh-worn monitors achieved acceptable overall balanced accuracies (≥80.0%) for SB, standing, and medium-to-vigorous PA classifications. In conclusion, it is advisable to position accelerometers on the thigh, collect MET data for ≥60 s, and ideally utilise population-specific trained algorithms.

## 1. Introduction

Sedentary behaviour (SB) and physical activity (PA) are independent modulators of healthy ageing [1,2,3,4] and/or prognosis following a health crisis [5], hence marking the importance of scrutinising the entire spectrum of physical behaviour. Such scrutiny is preferably quantified using 3D-accelerometry-based monitors rather than questionnaires to avoid the limitations of self-reporting [2,6,7]. Importantly, two key factors need to be considered to allow SB and/or PA dose–response studies vis à vis markers of health: good activity classification and accurate intensity estimation. The dose–response relationship between SB, PA, and markers of health reported in studies monitoring activity intensity with activity trackers has been used to update recommendations in physical activity guidelines [2], such as recommending light-intensity PA (LIPA) as a feasible long-term lifestyle choice in highly sedentary older people [8]. 

Accurately and objectively monitoring free-living intensity levels/movement behaviour is, however, challenging, especially in the elderly [9]. Firstly, most activity monitor algorithms have been designed for, and developed on, younger and healthier populations. Consequently, any established activity thresholds or cut-off points for activity intensities are unlikely to be applicable to other populations [2,10,11]. In addition, different decisions for the processing of data obtained from accelerometers do have an impact on the interpretation of PA and SB levels [12].

Ageing is associated with biomechanical, physiological, and metabolic characteristics that influence effort and the relative use of physiological reserves in carrying out activities of daily living [2,13,14,15]. Whilst for accurate activity intensity identification, individual physical and demographic characteristics should be included to allow for the development of tailored algorithms, which is not practical. As acceptable compromises, advances in the currently available commercial activity monitors would be to either utilise age-specific algorithms that incorporate the entire physical behaviour spectrum [2], or alternatively, universal algorithms to characterise and track specific activities such as sit-to-stand transitions [16,17].

Although there is an increasing number of studies within the literature on SB and PA dose–response effects on several health and quality-of-life outcomes in older persons [1,18,19,20], the data from the different laboratories are obtained with diverse monitors, with each using different algorithms [7,21,22,23,24,25]. Secondly, it appears that the anatomical site of monitor wear likely impacts on the apparatus’ ability to accurately detect posture and activity intensity. To draw a bigger picture of the distinct effects of SB and PA in the elderly, both the degree of monitor accuracy and agreement between monitors need to be established. This will enable researchers and end-users alike to pool all the information gathered from the numerous studies. In addition, where the output of a particular monitor markedly differs from that obtained by other units, this should be highlighted so that spurious cause–effect conclusions are avoided. Yet, an extensive comparison of activity monitors has rarely been conducted in the elderly, and evidence on their validity in older and slower moving people is limited [9,11,26].

The purpose of the current study was, therefore, to validate and compare six algorithms, four different activity monitors, on five anatomical placements for the quantification of activity intensities in older adults. This was conducted by (i) determining participant-specific and overall balanced accuracies per algorithm, (ii) comparing participant-specific balanced accuracies between algorithms, and (iii) benchmarking participant-specific and overall balanced accuracies per algorithm. It was hypothesised that wearing an activity monitor on the thigh, rather than the wrist, waist, or lower back, would enhance its ability to differentiate between standing and sitting/lying postures and increase the monitor’s accuracy in detecting physical activity intensity. It was also hypothesised that an algorithm specifically trained with data from older persons outperforms any other (proprietary) algorithm for each activity intensity when applied to older adults.

## 2. Materials and Methods

### 2.1. Participants

Twenty older adults (70.0 (12.0) years; 10 women) participated in this study. Exclusion criteria were: <60 years of age, not able to complete the laboratory-based activity protocol independently, any diagnosed neurological condition, being diabetic, terminally ill, or currently receiving cancer treatment, experiencing myocardial infarction in the previous 12 months or any currently unstable cardiovascular condition, any pulmonary disease or condition that would not allow for expired gas sampling, injuries or surgeries within the previous three months, previously advised by their physician not to undertake any physical activity/exercise, or not competent to make an informed decision about study participation.

This study was approved by the Medical Ethical Board of the University Hospital KU Leuven, Belgium (ESS-12.12.14(i)). All participants provided their written informed consent prior to participation.

### 2.2. Participant Characteristics

The demographics of the participants are reported in Table 1 and include age (years), sex (female/male), body height (to the nearest 0.1 cm; barefoot), body mass (to the nearest 0.1 kg; barefoot and light clothing only), body mass index (calculated by dividing body mass by squared body height (kg∙m^−2^)), then following a 20 min seated posture, resting oxygen consumption (RVO_2_; mL∙kg^−1^∙min^−1^; standard temperature and dry gas at standard barometric pressure; assessed while sitting quietly on a chair for four minutes (as this time frame does not affect the accuracy of energy expenditure measurements [27])), and resting heart rate (beats per minute) as a proxy for physical fitness levels [28], except for participants who used heart rate controlling medication. The self-selected walking speed on a treadmill was referred to as preferred walking speed (km∙h^−1^). Finally, a falls risk assessment tool (FRAT- [29]) classified risk of falling for each participant (low/medium/high).

### 2.3. Instrumentation

Four different activity monitors were simultaneously used for this study: ActiGraph wGT3X-BT (ActiGraph, Ft. Pensacola, FL, USA), ActivPAL3c VT (PAL Technologies, Glasgow, UK), GENEActiv Original (Activinsights Limited, Kimbolton, Cambridgeshire, UK), and DynaPort MM+ (McRoberts B.V., The Hague, The Netherlands). Each monitor was set to its default settings and worn as recommended by the manufacturer. 

ActiGraph wGT3X-BT (46 × 33 × 15 mm, 19 g) was sampled at the default setting of 30 Hz (with the low-frequency extension filter applied) and worn around the waist on the mid-axillary line of the right hip using an elastic band. ActivPAL3c VT (35 × 53 × 7 mm, 15 g) was sampled at 20 Hz and mounted on the right anterior mid-thigh (at 50% femur length, the latter being the distance between the trochanter major and the lateral femur epicondyle) using Tegaderm™ transparent film dressing (3M Health Care, St. Paul, MN, USA). GENEActiv Original (43 × 40 × 13 mm, 16 g) was worn on two locations, each having its own sampling frequency: non-dominant wrist using medical tape (100 Hz) and left anterior mid-thigh (at 50% femur length using Tegaderm™ transparent film dressing; 60 Hz). Finally, DynaPort MM+ (106.6 × 58 × 11.5 mm, 55 g) was worn on the middle of the lower back using an elastic band and was sampled at 100 Hz (Figure 1).

### 2.4. Indirect Calorimetry

A portable breath-by-breath metabolic system was used for indirect calorimetry (Oxycon Mobile JAEGER™/CareFusion, Hoechberg, Germany). The system comprised 2 units (sensor box and data exchange unit, each 126 × 96 × 41 mm) worn against the chest using a harness. In addition, a Polar T31 coded transmitter belt for heart rate monitoring (Polar Electro Oy, Kempele, Finland) and a face mask with a dead space of <30 mL (Hans Rudolph Inc, Kansas City, MO, USA) were used. A lightweight bi-directional 30 mL dead-space digital volume transducer (DVT) was connected to the facemask, to which a Nafion sampling tube for exhaled air was connected. Due to its low weight (950 g), the system caused minimal discomfort. Oxygen consumption (VO_2_), carbon dioxide production (VCO_2_), heart rate, respiratory rate, and tidal volume were measured continuously for the duration of the laboratory protocol. All measured data (gas and flow signals and heart rate) were sent telemetrically to a calibration and receiver unit, itself connected to a laptop (IBM, Armonk, NY, USA) where it was processed using JLAB (Carefusion Germany 234 GmbH, Hoechberg, Germany). All data were stored on an internal SD memory card inside the data exchange unit. The portable system was switched on at least 30 min prior to each participant’s arrival at the laboratory, and two-point gas calibration was completed using JLAB’s automated procedure.

### 2.5. Direct Observation

A GoPro Hero3 video camera (GoPro Inc., San Mateo, CA, USA) was attached to the front of the participant’s harness and used to record the entirety of the laboratory-based activity protocol. The recordings were stored on a microSD card and downloaded to a laptop after each session. All instrumentation was time-synchronised with a laptop, used for initialising the activity monitors and analysing the collected data.

### 2.6. Laboratory-Based Activity Protocol

Participants were instructed to refrain from physical exertion, stimulants, or smoking at least four hours prior to the tests to minimise interference with energy utilisation and thus produce reliable data [30]. The protocol consisted of 10 activities performed in a random order after a 20-min rest sitting quietly on a chair: (i) sitting while watching a TV broadcast, (ii) sweeping the floor, (iii) cycling on an ergometer (Technogym, Cesena, Italy), (iv) stair negotiations (walking up and down), (v) standing, (vi) walking with two shopping bags (2.5 kg each hand), (vii) walking on a treadmill at a self-selected speed (Forcelink, Culemborg, the Netherlands), (viii) sitting while doing desk work, (ix) doing the washing up, and (x) lying on a bed. All activities were performed for four minutes, where the first two minutes were used to reach a steady-state and the last two minutes were for data recording. The only exception to this was walking the stairs, as participants walked for two minutes before going up the stairs (one minute) and then walked for two minutes again before going down (one minute). Hence, the total duration of this activity was six minutes (2 + 1 + 2 + 1) instead of four. For data quality purposes, all activities were extended by a second at least to ensure activity continuation throughout the whole data recording period. Participants were instructed to perform each activity as naturally as possible and at their preferred pace. To prevent any fatigue carry-over effects, participants were seated between activities, and the next activity did not resume until their heart rate returned to resting level, as measured during the initial quiet sitting. The total duration of the activity protocol was approximately 60 min.

### 2.7. Algorithms

All activity monitors were analysed using their own (proprietary) algorithms and software, and the results were given per epoch, which varied for each monitor. ActiGraph wGT3X-BT was analysed in 60 s epochs using the Freedson Adult VM3 algorithm as provided in the ActiLife-software, version 6.13.3 (ActiGraph, Ft. Pensacola, FL, USA), which only differs from the newest version 6.13.5 released in October 2023 through not supporting the format for the CentrePoint Insight Watch. Data collected with ActivPAL3c VT were analysed in 15 s epochs using the ActivPAL3™-software, version 7.2.32 (PAL Technologies, Glasgow, UK). Two different algorithms were used to analyse the thigh-worn GENEActiv Original data. One algorithm is known as ‘Sedentary Sphere’ (thigh-worn version), and the data were analysed in 15 s epochs [22,25], while the other algorithm, the only custom-made algorithm specifically designed in older people [31], used Random Forest machine learning (100 trees) and 10 s epochs. Wrist-worn GENEActiv Original was also analysed in 15 s epochs but using a wrist-worn version of the ‘Sedentary Sphere’ algorithm [23,25]. Finally, DynaPort MM+ was analysed in 60 s epochs using the company’s online platform McRoberts version 2.2.1 (McRoberts B.V., The Hague, The Netherlands).

Oxygen consumption was measured by the Oxycon Mobile every 5 s. To determine the intensities of the activities performed during the protocol, VO_2_ per 5 s epoch was divided by the participant’s RVO_2_. This resulted in metabolic equivalent (MET) values. RVO_2_ was estimated by calculating the arithmetic mean over the 5 s epoch VO_2_ collected during the last two minutes while sitting quietly on a chair. Since MET values were calculated per 5 s epoch, this allowed for average MET values to be calculated for all intervals as used in the activity monitors, namely 10 s, 15 s, and 60 s epochs, respectively. The average MET values were used to classify activity intensities per epoch by first checking the MET value and then (if necessary, as different postures can produce similar MET values) the participant’s posture (Table 2). The classifications resulting from this scheme served as the criterion measure and were compared to the activity monitor outputs. To allow for a direct comparison with the criterion measure, if necessary, each epoch outcome per monitor was converted to these criterion measure classifications (Table 3).

Participant-specific confusion matrices were created to determine balanced accuracies per activity intensity for each monitor. In addition, overall confusion matrices per monitor were created by summing the participant-specific matrices. The balanced accuracies were calculated as the arithmetic mean of the sensitivity and specificity results per activity intensity for each monitor.
Sensitivity (%) = (True positives (N))/(True positives (N) + False negatives (N)) × 100(1)
Specificity (%) = (True negatives (N))/(True negatives (N) + False positives (N)) × 100(2)
where N represents the number of cases. Balanced accuracies of ≥80% were considered to be of an acceptable level [32].

### 2.8. MET Data Reliability by Epoch Length

Since MET values are a main part of the criterion measure classification scheme, it is important to check the reliability of this outcome for all epoch lengths used in the studied activity monitors, namely 10, 15, and 60 s, respectively. To achieve this, for each epoch length, a coefficient of variation (CV) per activity per participant [33,34] was calculated as:CV (%) = (SD_(activity/participant))/Arithmetic mean_(activity/participant) × 100 (3)
where SD represents standard deviation. Depending on data normal distribution, either the arithmetic mean (SD) or median (interquartile range (IQR)) was calculated over the moduli of all the CVs per epoch length to obtain sample-based reliability measures. A CV < 10% was considered acceptable. Additionally, CV consistency across the activity protocol was checked by examining the correlation between the CVs and accompanying MET values per epoch length. If a correlation was found, data dispersion was determined (SD or IQR).

### 2.9. Statistical Analyses

All data were checked for normality using the Shapiro–Wilk test. Participant characteristics are presented as the arithmetic mean (SD) (or median (IQR)). Balanced accuracies are reported as the arithmetic mean (95% confidence interval (95% CI) or median (95% CI)), except for those in the confusion matrices. To compare the balanced accuracies of the different monitors, a one-way repeated-measures ANOVA (or the Friedman test for non-parametric data) was performed. Where multiple post hoc comparisons were conducted, Bonferroni correction was applied to adjust *p*-values as per the equation below:Adjusted *p*-value_Bonferroni = *p*_value/k(4)
where k is the number of comparisons. *p*-values were considered statistically significant when *p* < 0.05. All statistical analyses were executed using IBM SPSS Statistics for Windows, version 23.0 (IBM Corp., Armonk, NY, USA).

## 3. Results

### 3.1. MET Data Reliability

MET CV values were negatively correlated with observed MET data for all epoch lengths, namely ρ −0.448 (*p* < 0.001) for 10 s epochs, ρ −0.482 (*p* < 0.001) for 15 s epochs, and ρ −0.236 (*p* = 0.001) for 60 s epochs (Figure 2), respectively. It is interesting to note that the CV was least in the 60 s epochs, suggesting that to obtain representative data, METs are best to be collected for ≥60 s. The IQRs of these epoch lengths’ CVs were between 7.9% and 19.8% (10 s), 6.5% and 16.7% (15 s), and 1.7% and 7.6% (60 s). For 10 s epochs, the sample-based CV was 12.1% (11.2%, 13.2%), while it was 10.7% (9.1%, 12.0%) for 15 s epochs and 3.3% (2.7%, 4.2%) for 60 s epochs. Overall, only the 60 s epoch CVs were <10%.

### 3.2. Overall Monitor Performance

The thigh-worn monitors (ActivPAL, Random Forest, and Sedentary Sphere—Thigh) showed the best performance in classifying sedentary behaviour (all balanced accuracies ≥ 94.2%, with sensitivity ≥ 99.3% and specificity ≥ 88.5%) (Table 4). The balanced accuracy of the other monitors (ActiGraph, DynaPort MM+, and GENEActiv Sedentary Sphere—Wrist) ranged between 73.6% and 75.5%. Their sensitivity values ranged between 67.2% and 85.7%, while specificity was between 65.4% and 80.1%.

Balanced accuracies for standing classification varied from 42.4% (DynaPort MM+) to 90.1% (GENEActive Sedentary Sphere—Thigh). The highest sensitivity was found for ActivPAL (94.0%) and the lowest for DynaPort MM+ (4.9%). Specificity was the highest for Random Forest (98.3%) and the lowest for DynaPort MM+ (79.8%).

ActiGraph showed the highest balanced accuracy for LIPA classification (69.7%), while DynaPort MM+ had the lowest (49.9%). Sensitivity values ranged from 0.0% (DynaPort MM+) to 66.7% (ActiGraph). Specificity was the highest for DynaPort MM+ (99.7%) and the lowest for ActiGraph (72.8%).

Finally, moderate-to-vigorous PA (MVPA) classification was between 68.8% (Sedentary Sphere—Wrist) and 85.4% (ActivPAL). Random Forest showed the highest sensitivity (83.7%), while ActiGraph had the lowest (40.4%). The specificity for MVPA ranged between 85.4% (Random Forest) and 98.4% (ActiGraph).

### 3.3. Monitor Comparison

The performance of sedentary classification was significantly different for ActivPAL and GENEActiv Random Forest when compared to all monitors, but not each other (Figure 3). Both showed higher participant-specific balanced accuracies. For classifying standing, GENEActiv Random Forest showed the most significant differences with other monitors, being lower than ActivPAL (−3.5%, −7.4%, −0.9%, *p* = 0.045) and higher than DynaPort MM+ (−55.8%, −58.8%, −54.6%, *p* < 0.001), respectively. GENEActiv Random Forest also showed most differences with monitors for LIPA classification. Participant-specific balanced accuracies in this monitor were higher than in ActivPAL (−9.7%, −14.3%, −5.0%, *p* < 0.001), DynaPort MM+ (−10.1%, −14.7%, −5.4%, *p* < 0.001), and GENEActiv Sedentary Sphere—Wrist (8.4%, 2.5%, −12.0%, *p* < 0.001). MVPA classification favoured ActivPAL and GENEActiv Random Forest, which had similar a performance and were significantly different to all monitors, except DynaPort MM+.

### 3.4. Monitor Benchmarking

Overall balanced accuracies for the classification of sedentary activity were only of an acceptable level (≥80.0%) in the thigh-worn monitors (ActivPAL, GENEActiv Random Forest, and Sedentary Sphere—Thigh) (Figure 4). Standing classification was acceptable in the same monitors, and also in GENEActiv Sedentary Sphere—Wrist. Interestingly, none of the monitors showed ≥80% overall balanced accuracy for classifying LIPA, but ActivPAL, DynaPort MM+, and GENEActiv Random Forest reached the ≥80% overall balanced accuracy threshold for MVPA classification.

The percentage of participants showing an acceptable level of participant-specific balanced accuracy revealed that the classification of sedentary activity was acceptable in all participants when using a thigh-worn monitor (Table 4). The other monitors showed a maximum of 52.6% only. Standing was classified acceptably in all participants when using ActivPAL or GENEActiv Sedentary Sphere—Thigh. In GENEActiv Random Forest and Sedentary Sphere—Wrist, this number was 85.0%, while it was 43.8% and 0.0% in ActiGraph and DynaPort MM+, respectively. Acceptable levels of LIPA classification were the highest in ActiGraph (33.3%), followed by GENEActiv Random Forest (5.0%). All other monitors failed to reach acceptable levels of LIPA classification. Acceptable MVPA classification varied significantly between the monitors. GENEActiv Random Forest displayed the highest degree of MVPA classification balanced accuracy (95.0%), followed by ActivPAL (85.0%) and DynaPort MM+ (80.0%). The remaining monitors only had acceptable levels in ≤40.0% of the participants: GENEActiv Sedentary Sphere—Thigh (40.0%), ActiGraph (33.3%), and GENEActiv Sedentary Sphere—Wrist (15.0%).

## 4. Discussion

As hypothesised, wearing accelerometers in anatomical positions that permit the clear identification of posture, combined with algorithms specially developed for older persons, was the most accurate for classifying activities and activity intensities in older adults. Indeed, the thigh-worn GENEActiv analysed with the Random Forest algorithm—developed in older people [31]—outperformed other algorithms/monitors in correctly identifying each activity intensity. Another notable observation was that longer epoch lengths proved more accurate than shorter ones to estimate intensity (MET value), suggesting that for accurate measures of physical activity, intensity measurements must be approximately 60 s.

Although most monitors showed good results for at least one activity intensity, across all activity intensities, ActivPAL is the only monitor with comparable performance to GENEActiv Random Forest trained on data from older adults. Interestingly, none of the monitors (including GENEActiv) showed acceptable outcomes (i.e., ≥80% overall balanced accuracy) for LIPA classification in our sample of older participants. This perhaps reflects the complexity of qualifying LIPA in this group and/or an inability for older individuals to consistently carry out an activity at that threshold. LIPA is suggested to be important for counteracting SB especially in older people based on positive health outcomes [8] as it minimises the need to engage in MVPA and hence may help maximise long-term compliance to adequate amounts of daily physical activity [35].

To check the potential cause for the low balanced accuracies for LIPA classification, the confusion matrix may be helpful. It shows both sensitivity and specificity values per monitor for each activity intensity, and it seems that sensitivity is the main issue. More specifically, three out of six algorithms (ActivPAL, DynaPort MM+, and GENEActiv Sedentary Sphere—Thigh) predominantly misclassified LIPA as standing, while two algorithms (GENEActiv Random Forest and Sedentary Sphere—Wrist) misclassified LIPA as MVPA. Interestingly, only ActiGraph did not have such an LIPA classification issue, potentially owing to its use of 60 s epochs and the associated acceptable MET CVs. It is noteworthy that MET CVs were quantified with the assumption that activities were performed in a metabolic steady-state and with matching biomechanics. Any discrepancy between these two factors could lead to inaccuracies. Since we found a negative correlation between CVs and METs, it may be that metabolic steady-state was not reached during lower activity intensities, such as standing or LIPA. 

This misclassification illustrates the reported challenge of activity monitoring in slower moving people, such as the elderly [9,11]. In normal ground walking, for instance, older persons tend to utilise a larger number, but smaller, steps for a given walking speed than younger people [36], which might result in smaller accelerations that are not much larger than the high sway that occurs during standing in older persons [37], contributing to the frequent misclassification of LIPA as standing. The fact that ActiGraph is the only monitor to use a low-frequency extension filter (30 Hz) might explain why it rarely misclassifies LIPA for standing, as such a filter helps to pick up slow movements, whereas other monitors (such as the three mentioned) do not sense it. LIPA misclassification may have also occurred due to the incorporation of household activities in our activity protocol. An activity such as washing dishes requires mainly upper limb action; hence, monitors not attached to this anatomical site will register less movement, while upper limb monitors might do the opposite.

Interestingly, GENEActiv Random Forest is the only non-upper-limb algorithm which misclassified LIPA with MVPA mostly. Presumably, this is caused by the fact that it is using pattern recognition, thereby causing the monitor to consider motion differently than just detecting the amount of movement. Thus, with the LIPA window being only small and yet exhibiting a similar pattern to MVPA, it is conceivable that misclassifications with MVPA could be made, as seen in a previous study that pertained to compare various methods of quantifying LIPA [38]. In addition, decisions during data processing on cut-offs for MVPA have been shown to affect the outcome [12]. 

Interestingly, a considerable amount of LIPA (≥15.7%), but MVPA in some cases too, was misclassified as sedentary activity. A plausible explanation comes from the cycling activity that was performed. For this activity, the posture, including thigh inclination near horizontal and hands holding the handlebars, potentially made classification difficult. Apart from helping to correctly classify cycling intensity, through thigh inclination, measuring thigh acceleration can also help to better distinguish between SB and standing [6,38]. As seen in our current data, the thigh-worn monitors performed better than the waist-worn ones (including lower back). Interestingly, wrist-worn monitors seem to handle these classifications better. This is important information for deciding what monitor is best to use if SB is a primary outcome measure.

Another consideration is what movement monitor epoch length to use for accurate activity classification and precise activity intensity. Notwithstanding differences in monitor placement and sampling frequency, our data showed better performance (i.e., high balanced accuracies, sensitivity, and specificity) with shorter epoch lengths, which is in line with the findings of previous research [39]. However, the MET CVs suggest otherwise. The smallest and only acceptable sample-based MET CV was found for 60 s epochs, while the largest were found for 10 s epochs. Despite this, monitor performance (i.e., balanced accuracies, sensitivity, and specificity) was better with the smaller epoch lengths. Since the set activities were performed in the same fashion throughout the whole laboratory-based protocol, we would suggest that better performance in epochs with higher MET CVs is not a direct result of smart or robust algorithms. Instead, because MET CVs were calculated from MET values, which were converted into intensities and eventually cross-validated, it rather proves the robustness of the classification scheme.

The main explanation for GENEActiv Random Forest to outperform the other monitors for the classification of activity types may be its use of pattern recognition instead of cut-off points. With most of the studied algorithms being proprietary, their exact mathematical iterations are unclear. However, it is safe to assume that they would largely rely on cut-off points. Other studies have also shown that machine learning is more accurate than cut-off points in activity monitoring [40,41], and, consequently, pattern recognition has been suggested as the future standard [7]. Nevertheless, most current studies are still using cut-off point algorithms, potentially as these are more straight-forward to apply, even for the non-mathematically minded [7]. Machine learning algorithms supposedly make the requirement of specific anatomical attachment sites of an activity monitor less relevant [41]. However, we have shown here that thigh-mounted monitors outperform monitors placed elsewhere, especially for the activity classification of standing vs. sedentary behaviours, suggesting that anatomical position is important, even when using machine learning algorithms. Our findings also lend further support to ActivPAL as a good performing monitor and algorithm (both for activity classification and intensity), and its widespread use as a criterion measure to validate other monitors [6,42].

Contextualising our findings in the light of the existing literature is challenging, not least because of the scarcity of comparable ‘mobility monitor’ validation studies in older adults and the use of different outcome measures and/or different sampling frequencies. Nevertheless, comparisons with prior studies, which applied the monitors in the same fashion (none performed in older adults specifically, except for DynaPort MM+), show that the results of the ActivPAL monitor in our current study were comparable in the classification of SB (97.4% vs. lying horizontal 100.0% and sitting 91.0%), but worse for upright activities, such as standing and stepping (≤87.5% vs. 99.0%), to that reported previously [22,43]. As for ActiGraph, our results for sedentary activity were slightly better than the accuracy obtained in a previous study (≤72.0% in theirs compared to 74.2% in ours), while the accuracy of detecting upright activities was slightly lower (≤69.7% vs. 74.0%) [22]. However, Kerr et al. [21] using GT3X+ ActiGraph accelerometer sampling at 30 Hz (ActiLife v6.2.1 software; 60–s epochs) showed worse mobility detection accuracy for all activities (≤43.0% vs. ≥69.4%) except sitting (84.0% vs. 74.2%). The accuracy of the DynaPort MM+ monitor in our current study was marginally lower than the results found by Hollewand et al. [24]. They showed 79.6% for lying, 87.6% for sitting (both vs. 75.5%), 81.5% for standing (vs. 42.4%), and 91.7% for locomotion (vs. ≥49.9%). A study by Rowlands et al. [23] found accuracies of 74.0% and 91.0% for classifying SB and upright activities when using GENEActiv Sedentary Sphere—Wrist. The results in our current study are similar for sedentary activity (73.6%) but worse for standing (81.5%), LIPA (53.9%), and MVPA (68.8%). When comparing accuracies of the GENEActiv Sedentary Sphere—Thigh results from our current study with those by Edwardson et al. [22], their accuracies were better, ≥99.8% vs. 94.2% for SB and ≥88.3% vs. ≥59.0%, respectively, for upright activities, potentially owing to these previous authors using a higher data acquisition frequency (100 Hz vs. 60 Hz) and a younger (27 ± 6 years vs. 70 ± 12 years) and lower BMI (24 ± 4 kg∙m^−2^ vs. 27 ± 4 kg∙m^−2^) study sample. The impact of the age discrepancy especially between our study and previous ones cannot be underestimated.

To our knowledge, we are the first group to have validated the machine learning technique random forest for thigh-worn accelerometry [31]. Here, we show that the present results are comparable to those seen in our previous work [31], i.e., 99.6% vs. 97.0% for SB, 95.5% vs. 84.3% for standing, 80.6% vs. 61.0% for LIPA, and 95.1% vs. 84.5% for MVPA for the previous and our current study, respectively.

### 4.1. Study Limitations and Strengths

The fact that this study was performed in a laboratory setting is a limitation because it does not show any information on how well the monitors will perform during free-living. We did ask the participants, however, to perform the activities as naturally as possible. In addition, we only explored the efficacy of detecting 10 activities, while in daily free-living conditions, many more activities are performed. Despite these shortcomings, this study does provide useful information on how monitors will perform compared to each other. One of the strengths of the current study is that we concurrently compared a good selection of activity monitors used in research. Moreover, we used these as recommended by their developers/manufacturers, including the optimal body location, epoch length, and software/open-source algorithm for analysis.

Overall, generalisation of our findings is difficult because we only used a small study sample (N = 20) of fit and healthy older adults. Nevertheless, the current study presents highly valuable and important insights for activity monitoring in an understudied age group. 

### 4.2. Recommendations for Future Studies

Future research should validate and compare the studied monitors for quantifying free-living (rather than laboratory-based) physical activity levels in the elderly. We would also recommend that device improvements be made in terms of ability to accurately detect LIPA, especially at least in older adults. 

Furthermore, future studies should compare the performance of the various activity monitors in a wider range of ages, from young to more senior populations. Additionally, it could be explored to what extent an improved detection of physical activity levels and sedentary behaviour may enhance optimal physical behaviour in hospitalised patients [26]. Finally, the data-driven algorithms also show the potential to provide individualised advice for rehabilitation; thus, they fit with the potency of data-driven healthcare to improve general health [44].

## 5. Conclusions

It appears that thigh-worn monitors provide better information about physical activity type and intensity than other anatomical locations. In addition, the algorithm employed is important, where a pattern detector, rather than a cut-off algorithm, appears to provide the best results. It should also be considered that short epoch lengths produce better algorithm performance (i.e., high balanced accuracies, sensitivity, and specificity); therefore, they are better suited for activity classification. On the other hand, a longer epoch length is ideal for estimating activity intensity (MET values), so it is recommended to have a recording period of ~60 s for this aspect of physical behaviour analysis.

## Figures and Tables

**Figure 1 sensors-24-00895-f001:**
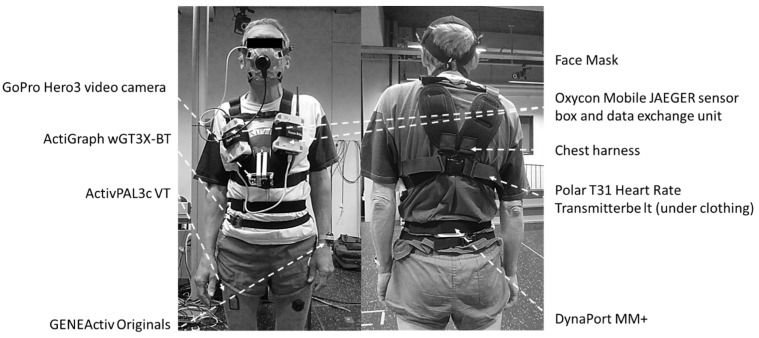
Study participant wearing all activity monitors and laboratory protocol peripherals.

**Figure 2 sensors-24-00895-f002:**
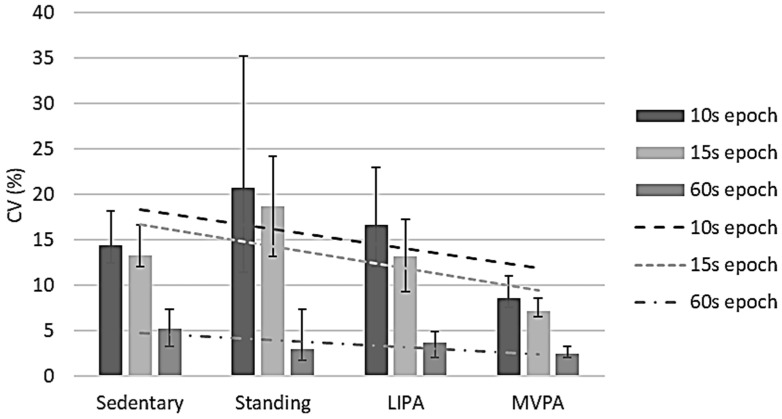
Metabolic equivalent value reliability per activity intensity per epoch length. CV, coefficient of variation; LIPA, light-intensity physical activity; MVPA, moderate-to-vigorous physical activity. Error bars represent 95% confidence intervals. Dashed lines show correlations between coefficients of variation and intensities per epoch length.

**Figure 3 sensors-24-00895-f003:**
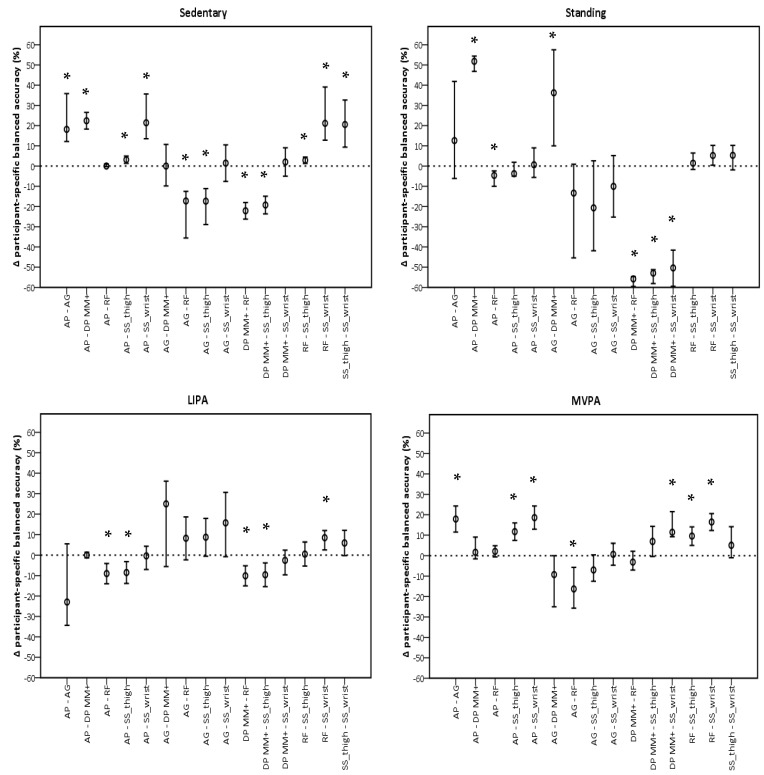
Pairwise comparisons between monitors per intensity using participant-specific balanced accuracies. AP, ActivPAL; AG, ActiGraph; DP MM+, DynaPort MM+; RF, Random Forest; SS_thigh, Sedentary Sphere—Thigh; SS_wrist, Sedentary Sphere—Wrist; LIPA, light-intensity physical activity; MVPA, moderate-to-vigorous physical activity. Error bars represent 95% confidence intervals; dashed line represents no difference; * *p* < 0.05.

**Figure 4 sensors-24-00895-f004:**
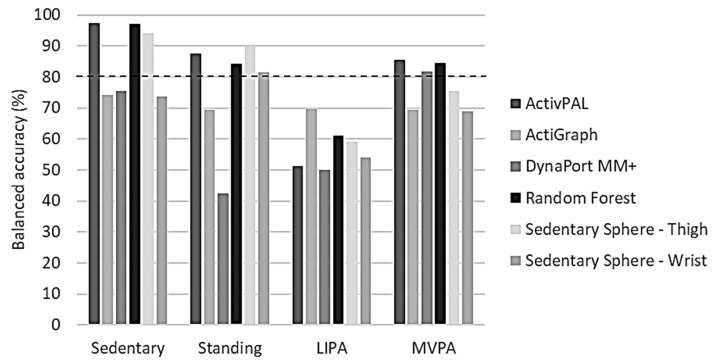
Benchmarking of overall balanced accuracies per activity intensity for each tested algorithm. LIPA, light-intensity physical activity; MVPA, moderate-to-vigorous physical activity. Dashed line represents threshold for acceptable algorithm performance (80%).

**Table 1 sensors-24-00895-t001:** Study sample characteristics. BMI, body mass index; RVO_2_, resting oxygen consumption. * Only determined for participants not taking any heart rate controlling medication. ^¶^ Values represent either arithmetic mean (standard deviation) or median (interquartile range).

Age (years)	70.0 (12.0) ^¶^	
Sex	10 Women	10 Men
Body mass (kg)	73.4 (13.0)	
Body height (cm)	165.6 (8.1)	
BMI (kg∙m^−2^)	26.7 (3.6)	
RVO_2_ (ml∙kg^−1^∙min^−1^)	2.87 (0.52)	
Physical fitness level *	3 Less than good	11 Good or better
Preferred walking speed (km∙h^−1^)	2.6 (2.0) ^¶^	
Falls risk	19 Low	1 Medium or high

**Table 2 sensors-24-00895-t002:** Criterion measure classification scheme. MET, metabolic equivalent; LIPA, light-intensity physical activity; MVPA, moderate-to-vigorous physical activity.

Rules	Intensity Classification
1. If MET ≤ 1.5 and posture = sedentary, then	Sedentary
2. Else: If MET ≤ 1.5 and posture ≠ sedentary, then	Standing
3. Else: If MET > 1.5 and <3, then	LIPA
4. Else: If MET ≥ 3, then	MVPA

**Table 3 sensors-24-00895-t003:** Monitor classification conversion scheme. MET, metabolic equivalent; LIPA, light-intensity physical activity; MVPA, moderate-to-vigorous physical activity; VM, vector magnitude. N/a, Not applicable.

Rules	Classification
ActivPAL	
If epoch time predominantly = Sedentary, then	Sedentary
Else: If epoch time predominantly = Upright, then	Standing
Else: If epoch time predominantly = Stepping and MET < 3, then	LIPA
Else: If epoch time predominantly = Stepping and MET ≥ 3, then	MVPA
ActiGraph	
If epoch time predominantly = Sitting or Lying, then	Sedentary
Else: If epoch time predominantly = Standing and VM = 0, then	Standing
Else: If epoch time predominantly = Standing and VM < 2690, then	LIPA
Else: If epoch time predominantly = Standing and VM ≥ 2690, then	MVPA
DynaPort MM+	
If epoch class = Sitting or Lying, then	Sedentary
Else: If epoch class = Standing, then	Standing
Else: If epoch class = Shuffling or Walking and MET < 3, then	LIPA
Else: If epoch class = Shuffling or Walking and MET ≥ 3, then	MVPA
GENEActiv Original—Thigh—Random Forest	
Classifications of this monitor are in line with the criterion measure	N/a
GENEActiv Original—Thigh and Wrist—Sedentary Sphere	
If epoch intensity/activity = Sleep, then	Sedentary
Else: If epoch intensity/activity = Sedentary or Light and posture = Sit/lie, then	Sedentary
Else: If epoch intensity/activity = Sedentary and posture = Standing, then	Standing
Else: If epoch intensity/activity = Light and posture = Standing, then	LIPA
Else: If epoch intensity/activity = Moderate or Vigorous, then	MVPA

**Table 4 sensors-24-00895-t004:** Algorithm cross-validation confusion matrix. LIPA, light-intensity physical activity; MVPA, moderate-to-vigorous physical activity. Bold values represent the number of correct classifications.

Monitor	Intensity	Reference				Sensitivity (%)	Specificity (%)	Balanced Accuracy (%)	Acceptable Level (%)
		Sedentary	Standing	LIPA	MVPA				
ActivPAL	Sedentary	**563**	0	53	0	99.3	95.4	97.4	100.0
	Standing	4	**156**	192	102	94.0	80.9	87.5	100.0
	LIPA	0	0	**17**	37	5.0	97.3	51.2	0.0
	MVPA	0	10	76	**519**	78.9	92.0	85.4	85.0
ActiGraph	Sedentary	**95**	11	21	22	68.3	80.0	74.2	52.6
	Standing	8	**16**	0	0	41.0	97.8	69.4	43.8
	LIPA	8	12	**50**	71	66.7	72.8	69.7	33.3
	MVPA	0	0	4	**63**	40.4	98.4	69.4	33.3
DynaPort MM+	Sedentary	**126**	37	27	35	85.7	65.4	75.5	40.0
	Standing	21	**2**	40	18	4.9	79.8	42.4	0.0
	LIPA	0	0	**0**	1	0.0	99.7	49.9	0.0
	MVPA	0	2	10	**114**	67.9	95.5	81.7	80.0
Random Forest	Sedentary	**842**	0	103	1	100.0	94.1	97.0	100.0
	Standing	0	**173**	37	4	70.3	98.3	84.3	85.0
	LIPA	0	45	**160**	159	31.7	90.3	61.0	5.0
	MVPA	0	28	205	**841**	83.7	85.4	84.5	95.0
Sedentary Sphere—Thigh	Sedentary	**566**	5	92	37	99.8	88.5	94.2	100.0
	Standing	0	**149**	97	53	89.8	90.4	90.1	100.0
	LIPA	1	12	**116**	215	34.3	83.6	59.0	0.0
	MVPA	0	0	33	**356**	53.9	96.9	75.4	40.0
Sedentary Sphere—Wrist	Sedentary	**381**	17	111	104	67.2	80.1	73.6	40.0
	Standing	178	**131**	31	40	78.9	84.1	81.5	85.0
	LIPA	8	13	**78**	193	23.1	84.6	53.9	0.0
	MVPA	0	5	118	**324**	49.0	88.5	68.8	15.0

## Data Availability

The raw data supporting the conclusions of this article will be made available by the authors on request.

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
