# Peer review of "Concurrent Validity of Four Activity Monitors in Older Adults"

_sensors, 2024, doi:10.3390/s24030895_

Round 1

Reviewer 1 Report

Comments and Suggestions for Authors

-Addresses a timely and critical issue: Accurate activity monitoring in older adults is crucial for promoting healthy aging, and investigating placement and algorithm optimization is valuable.

- The study simultaneously compares four prominent activity monitors on ten different activities, providing a robust comparison.

- With only 20 participants, generalizability of the findings might be limited.

- While ten activities were included, they may not fully represent the daily lifestyle variations of older adults.

-Testing the trained random forest algorithm on new data for further validation would strengthen its credibility.

-Increase the sample size to enhance generalizability of the findings.

-Include a wider range of activities that reflect the diversity of older adults' daily routines.

-Validate the trained random forest algorithm on an independent dataset to assess its external validity.

-Analyze the performance of monitors and algorithms across different age groups within the older adult population.

-Provide more details about the training data used for the random forest algorithm, including its size, diversity, and potential biases.

-The paper should be enriched with references to existing research papers on The Data-Driven Future of Healthcare such as https://doi.org/10.58496/MJBD/2023/010 This would strengthen the argument and provide readers with additional resources for further exploration.

-To enhance the paper's impact, the author should discuss potential future directions for this research.

Author Response

R1Q1-Addresses a timely and critical issue: Accurate activity monitoring in older adults is crucial for promoting healthy aging, and investigating placement and algorithm optimization is valuable.

- The study simultaneously compares four prominent activity monitors on ten different activities, providing a robust comparison.

R1R1: We thank the reviewer for these positive comments on our manuscript.

R1Q2- With only 20 participants, generalizability of the findings might be limited.

R1R2: We agree that a large number of participants is often useful. The key message in this paper however centres around contrasting how different commercially available accelerometers, on anatomical sites recommended by each provider, perform relative to one another. Thus, we propose that the message is impactful as the data are clear-cut, and we do not expect that adding any more participants would have significantly changed the outcome. The robustness of our observations is assured associating the energy demands of each task, and simultaneous accelerometers recording from 5 anatomical sites, analysed 6 ways.

R1Q3- While ten activities were included, they may not fully represent the daily lifestyle variations of older adults.

R1R3: The 10 activities are laboratory-based and not free-living moving, as the idea is to compare how in different movement, each of the 5 accelerometer performs. Activities were selected so that they span the array of physical behaviours from sedentary to medium-to-vigorous physical activity, in this age group.  Thus, they represent different accelerations for the monitors to record. The point was not so much to represent daily lifestyle variations but rather the ability of the different sensors and sensor positions to detect movement and the associated activity levels in terms of low, medium, or high intensity.

R1Q4-Testing the trained random forest algorithm on new data for further validation would strengthen its credibility.

R1R4: We have indeed previously trained and validated the random forest algorithm in one of our previous publications. (Wullems et al, 2017) in a sample of 40 similarly older people.

Ref: Wullems JA, Verschueren SMP, Degens H, Morse CI and Onambélé GL. Performance of thigh-mounted triaxial accelerometer algorithms in objective quantification of sedentary behaviour and physical activity in older adults. PLoS One 2017;12:e0188215

R1Q5-Increase the sample size to enhance generalizability of the findings.

R1R5:   Please refer to R1R2.

R1Q6-Include a wider range of activities that reflect the diversity of older adults' daily routines.

-Validate the trained random forest algorithm on an independent dataset to assess its external validity.

R1R6: Please refer to R1R3 and R1R4.

R1Q7-Analyze the performance of monitors and algorithms across different age groups within the older adult population.

R1R7: This would be interesting also indeed. We had stated within at the end of the discussion that  ‘Future research should validate and compare the studied monitors for quantifying free-living (rather than laboratory based) physical activity levels in the elderly’. (see discussion lines 503-505).

We have now expanded this further by adding that ‘Furthermore, future studies should compare the performance of the various activity monitors in a wider range of ages, from young to more senior populations’.

R1Q8-Provide more details about the training data used for the random forest algorithm, including its size, diversity, and potential biases.

R1R8: Please note as stated in R1RA that the validation of the Random Forest was not the topic of the current paper but its comparison relative to other accelerometers and algorithms, all algorithms used, were taken here at face value. We previously validated the Random forest algorithm in one of our previous publications (Wullems et al, 2017). The other algorithms also have been previously validated by previous authors or the manufacturers. To go into the validation protocol of each of the algorithms in the current paper would detract from the focus of the paper. We have opted to refer the reader back to the papers where this is done.

Ref: Wullems JA, Verschueren SMP, Degens H, Morse CI and Onambélé GL. Performance of thigh-mounted triaxial accelerometer algorithms in objective quantification of sedentary behaviour and physical activity in older adults. PLoS One 2017;12:e0188215

R1Q9-The paper should be enriched with references to existing research papers on The Data-Driven Future of Healthcare such as https://doi.org/10.58496/MJBD/2023/010 This would strengthen the argument and provide readers with additional resources for further exploration.

R1R9: Thank you for this comment. We have added some more recent references to the Introduction and looked at the suggested reference and included it as follows (lines 506-511):

Additionally, it could be explored to what extent an improved detection of physical activity levels and sedentary behaviour may enhance optimal physical behaviour in hospitalised patients (Becker et al., 2023). Finally, the data-driven algorithms also show the potential to provide individualised advice for rehabilitation, thus fits with the potency of data-driven healthcare to improve general health (Amri & Abed, 2023).

R1Q10-To enhance the paper's impact, the author should discuss potential future directions for this research.

R1R10: Thank you for this suggestion. We have now clearly delineated (through a section title) and expanded on, a ‘future studies recommendation’ paragraph (see lines 504-511)included in the Discussion.

Reviewer 2 Report

Comments and Suggestions for Authors

This study makes a good effort in trying to validate the machine learning technique Random Forest for thigh-worn accelerometry. It is a very interesting study, but there are some revisions to be made.

1. In Introduction section, the authors should update the previous studies and references. There are more recent studies, as the following: - Longhini J, Marzaro C, Bargeri S, Palese A, Dell'Isola A, Turolla A, Pillastrini P, Battista S, Castellini G, Cook C, Gianola S, Rossettini G. Wearable Devices to Improve Physical Activity and Reduce Sedentary Behaviour: An Umbrella Review. Sports Med Open. 2024 Jan 14;10(1):9. - Kraaijkamp JJM, Stijntjes M, De Groot JH, Chavannes NH, Achterberg WP, van Dam van Isselt EF. Movement Patterns in Older Adults Recovering From Hip Fracture. J Aging Phys Act. 2024 Jan 12:1-9. - Eke H, Bonn SE, Trolle Lagerros Y. Wrist-worn accelerometers: Influence of decisions during data collection and processing: A cross-sectional study. Health Sci Rep. 2024 Jan 10;7(1):e1810. - Becker ML, Hurkmans HLP, Verhaar JAN, Bussmann JBJ. Validation of the Activ8 Activity Monitor for Monitoring Postures, Motions, Transfers, and Steps of Hospitalized Patients. Sensors (Basel). 2023 Dec 28;24(1):180. - Annear M, Kidokoro T, Shimizu Y. Physical Activity and Health of Middle-Aged and Older Japanese Across the COVID-19 Pandemic: Differential Outcomes Highlight a Problematic Life Stage. J Aging Phys Act. 2024 Jan 8:1-12.

(other publications in 2022 and 2023 have also been made).

2. In the Material and Methods, the authors should enter the protocol number regarding the approval of the ethics committee.

Comments on the Quality of English Language

Minor editing of English language required.

Author Response

R2Q1-This study makes a good effort in trying to validate the machine learning technique Random Forest for thigh-worn accelerometry. It is a very interesting study, but there are some revisions to be made.

R2R1: We thank our reviewer for these positive comments on our manuscript.

R2Q2- In Introduction section, the authors should update the previous studies and references. There are more recent studies, as the following:

- Longhini J, Marzaro C, Bargeri S, Palese A, Dell'Isola A, Turolla A, Pillastrini P, Battista S, Castellini G, Cook C, Gianola S, Rossettini G. Wearable Devices to Improve Physical Activity and Reduce Sedentary Behaviour: An Umbrella Review. Sports Med Open. 2024 Jan 14;10(1):9.

- Kraaijkamp JJM, Stijntjes M, De Groot JH, Chavannes NH, Achterberg WP, van Dam van Isselt EF. Movement Patterns in Older Adults Recovering From Hip Fracture. J Aging Phys Act. 2024 Jan 12:1-9.

- Eke H, Bonn SE, Trolle Lagerros Y. Wrist-worn accelerometers: Influence of decisions during data collection and processing: A cross-sectional study. Health Sci Rep. 2024 Jan 10;7(1):e1810.

- Becker ML, Hurkmans HLP, Verhaar JAN, Bussmann JBJ. Validation of the Activ8 Activity Monitor for Monitoring Postures, Motions, Transfers, and Steps of Hospitalized Patients. Sensors (Basel). 2023 Dec 28;24(1):180.

- Annear M, Kidokoro T, Shimizu Y. Physical Activity and Health of Middle-Aged and Older Japanese Across the COVID-19 Pandemic: Differential Outcomes Highlight a Problematic Life Stage. J Aging Phys Act. 2024 Jan 8:1-12.

(other publications in 2022 and 2023 have also been made).

R2R2: We thank our reviewer for alerting us to these recent papers. We have now incorporated 5/5 of the recommended articles in the revised Introduction and Discussion.

 R2Q3. In the Material and Methods, the authors should enter the protocol number regarding the approval of the ethics committee.

R2R3: We have now added the protocol number of the ethics committee to the manuscript (under Participants). This is ESS-12.12.14(i).

R2Q4. Minor editing of English language required.

R2R4: We had now systematically combed through the manuscript and attended to all necessary minor edits.